# Evolution of Asian Corn Borer Resistance to Bt Toxins Used Singly or in Pairs

**DOI:** 10.3390/toxins11080461

**Published:** 2019-08-06

**Authors:** Yueqin Wang, Yudong Quan, Jing Yang, Changlong Shu, Zhenying Wang, Jie Zhang, Angharad M. R. Gatehouse, Bruce E. Tabashnik, Kanglai He

**Affiliations:** 1The State Key Laboratory for Biology of Plant Diseases and Insect Pests, Institute of Plant Protection, Chinese Academy of Agricultural Sciences, Beijing 100193, China; 2School of Natural and Environmental Sciences, University of Newcastle, Newcastle upon Tyne NE1 7RU, UK; 3Department of Entomology, University of Arizona, Tucson, AZ 85721, USA

**Keywords:** *Bacillus thuringiensis*, cry toxin, pyramid strategy, resistance, *Ostrinia furnacalis*

## Abstract

Transgenic crops producing insecticidal proteins from *Bacillus thuringiensis* (Bt) have revolutionized pest control, but the benefits of this approach have been reduced by the evolution of resistance in pests. The widely adopted ’pyramid strategy’ for delaying resistance entails transgenic crops producing two or more distinct toxins that kill the same pest. The limited experimental evidence supporting this strategy comes primarily from a model system under ideal conditions. Here we tested the pyramid strategy under nearly worst-case conditions, including some cross-resistance between the toxins in the pyramid. In a laboratory selection experiment with an artificial diet, we used Bt toxins Cry1Ab, Cry1F, and Cry1Ie singly or in pairs against *Ostrinia furnacalis*, one of the most destructive pests of corn in Asia. Under the conditions evaluated, pairs of toxins did not consistently delay the evolution of resistance relative to single toxins.

## 1. Introduction

Genetically engineered crops that produce insecticidal proteins from the bacterium *Bacillus thuringiensis* (Bt) have revolutionized pest control and were planted on over 100 million hectares worldwide in 2017 [1]. These transgenic Bt crops can suppress pests, reduce insecticide use, and increase farmer profits [2,3,4,5,6]. However, adaptation of insect pests to Bt toxins has reduced these benefits [7,8,9,10].

The ‘pyramid strategy’, which entails transgenic crops producing two or more distinct toxins that kill the same pest, has been widely adopted to delay evolution of pest resistance to Bt crops [11]. Of the eight conditions favoring durability of Bt crop pyramids, five are especially important: (1) Each toxin in the pyramid can kill all or nearly all susceptible insects; (2) resistance is inherited as a recessive trait; (3) no cross-resistance occurs between toxins in the pyramid; (4) refuges of non-Bt host plants that allow survival of susceptible insects are sufficiently abundant; and (5) pyramids are not grown concurrently with single-toxin plants that produce one of the toxins in the pyramid [11]. Although the pyramid strategy has been studied extensively with theoretical models, experimental data from selection experiments testing toxins singly and in combinations are limited [11,12,13,14,15,16]. Moreover, previous selection experiments have focused on ideal or nearly ideal scenarios, which may not reflect field conditions.

The primary experimental evidence supporting the use of the pyramid strategy for Bt crops is from a model system with the diamondback moth and noncommercial Bt broccoli plants that produce Bt toxins Cry1Ac and Cry1C [15,16]. In the initial study with this model system, the experimental conditions included all five favorable factors listed above [16]. A second study with this system evaluated condition (5) and showed that concurrent exposure to pyramids and single-toxin plants accelerated resistance [15]. Unlike the nearly ideal conditions of the model system, review of the relevant data indicates that in many cases, each toxin in a pyramid does not kill nearly all susceptible insects, resistance is not recessive, and some cross-resistance occurs between the toxins in these pyramids [11,14]. When Bt toxins are first used in single-toxin crops, they typically cause >90% mortality of susceptible target pests. However, pyramids are often deployed after field-evolved resistance has decreased the efficacy of one or more toxins in the pyramid, so the mortality caused by each toxin alone may be 50% or less [9,10,11]. Furthermore, refuge abundance is often limited because of the lack of compliance with regulations, lack of regulations requiring abundant refuges, or both [10]. Modeling results suggest the increased durability of pyramids relative to single-toxin plants will be greatly diminished under such suboptimal conditions [12,17,18]. However, previous studies have not tested this hypothesis experimentally.

Here we tested the durability of pyramids of two Bt toxins versus single Bt toxins under nearly worst-case conditions using laboratory selection experiments with the Asian corn borer (*Ostrinia furnacalis*), one of the most damaging crop pests in Asia [19,20,21,22]. We tested Bt toxins Cry1Ab, Cry1F, and Cry1Ie alone and in pairs (Cry1Ab + Cry1F and Cry1Ab + Cry1Ie) incorporated in an artificial diet. Some widely planted Bt corn pyramids produce Cry1Ab and Cry1F [11]. Cry11e, which is not produced by currently commercialized crops, has been proposed for use in pyramids based on previous work suggesting little or no cross-resistance occurs between Cry1Ie and either Cry1A toxins or Cry1F in *O. furnacalis* or in *Ostrinia nubilalis*, an important corn pest in the United States [23,24,25,26,27,28]. However, moderate cross-resistance has been reported between Cry1Ab and Cry1F in *O. furnacalis* [25,27]. Moreover, in field efficacy tests, Bt corn producing Cry1Ab, Cry1F or Cry1Ie alone or Cry1Ie + Cry1Ac (which is similar to Cry1Ab) provided significant protection but did not meet the high-dose standard against *O. furnacalis* [10,23,29]. In this study, unlike previous work, the first four favorable conditions listed above did not apply. The mortality caused by each toxin alone was at most 50%, resistance was not recessive, some cross-resistance occurred between the toxins, and refuges were absent. These unfavorable conditions may reflect some aspects of field situations, but do not necessarily represent conditions for currently available Bt corn products against this pest. Under the conditions evaluated, the pyramids tested did not consistently delay the evolution of resistance relative to single Bt toxins.

## 2. Results

We tracked the evolution of resistance using the resistance ratio (RR), which is the toxin concentration killing 50% of larvae (LC_50_) for a selected strain divided by the LC_50_ of the same toxin for the susceptible parent strain (S) from which the selected strains were derived. After 14 generations of selection, the RR for Bt toxins Cry1Ab, Cry1F, and Cry1Ie was not lower in the strains selected with pyramids versus single toxins in any of the four pairwise comparisons (Figure 1 and Appendix A).

After 14 generations, the RR for Cry1Ab was 28 (95% CI: 23–35), 32 (23–45), and 60 (47–78) for the strains selected with Cry1Ab alone, Cry1Ab + Cry1F and Cry1Ab + Cry1Ie, respectively (Figure 1a,b). By the conservative criterion of no overlap between the 95% CIs, the RR for Cry1Ab after 14 generations did not differ significantly between the strains selected with Cry1Ab versus Cry1Ab + Cry1F, but it was significantly higher for the strain selected with Cry1Ab + Cry11e than either of the other two strains.

For Cry1F and Cry1Ie, RRs after 14 generations of selection were >800 in all of the single-toxin and pyramid treatments (Figure 1c,d). After 14 generations, resistance to these two toxins was so high that the highest concentrations tested did not kill 50% of the larvae and we could not accurately estimate the LC_50_ values and RRs. The last generation that yielded RRs with 95% CIs for both the pyramid and single-toxin treatments was generation 6 for Cry1F and generation 5 for Cry1Ie (Figure 1c,d). At generation 6, the resistance ratio for Cry1F was 560 (420–740) for the strain selected with Cry1F alone versus 240 (170–340) for the strain selected with Cry1Ab + Cry1F (Figure 1c). This is one of the few comparisons where the resistance ratio was significantly lower for the pyramid than the single-toxin treatment. At generation 5, the RR for Cry1Ie did not differ significantly between the strains selected with Cry1Ie alone (55 (42–72)) versus Cry1Ab + Cry1Ie (85 (63–116)) (Figure 1d).

For each of the three strains selected with a single toxin, we evaluated cross-resistance to the other two toxins after 29 generations of selection (Table 1). The selection with Cry1Ab caused significant 1.6-fold resistance to Cry1F, and selection with Cry1F caused significant 2.4-fold resistance to Cry1Ab (Table 1). In contrast with this symmetrical cross-resistance between Cry1Ab and Cry1F, we found asymmetrical cross-resistance between Cry1Ie and the other two toxins. Selection with Cry1Ab or Cry1F did not significantly affect susceptibility to Cry1Ie, but selection with Cry1Ie caused significant 2.5-fold cross-resistance to Cry1Ab and >160-fold cross-resistance to Cry1F (Table 1).

## 3. Discussion

Consistent with predictions from theory [12,17,18], under the nearly worst-case conditions examined here, evolution of resistance by *O. furnacalis* was not consistently delayed by pyramids of two toxins relative to selection with single toxins. Contrary to conditions favoring durability of pyramids, the mortality caused by each toxin alone was at most 50%, resistance was not recessive, refuges were absent, and some cross-resistance occurred between the toxins. The durability of pyramids was not consistently less than that of single toxins, confirming predictions from theory [12,17,18]. Although insects were not exposed concurrently to single toxins and pyramids, the other unfavorable conditions in this study approached a worst-case scenario for pyramids, particularly when compared with the nearly ideal scenarios tested previously [15,16].

The significant, symmetrical cross-resistance between Cry1Ab and Cry1F seen here of 1.6- and 2.4-fold is low relative to the 15-fold resistance to Cry1Ab caused by selection with Cry1Ab and the >1600-fold resistance to Cry1F caused by selection with Cry1F (Table 1). In addition, the cross-resistance between Cry1Ab and Cry1F in this study is somewhat lower than in previous studies of *O. furnacalis*. In previous work, selection with Cry1Ab yielded 40-fold resistance to Cry1Ab and 6-fold cross-resistance to Cry1F [27]. Previous selection with Cry1F yielded >1700-fold resistance to Cry1F and 23-fold cross-resistance to Cry1Ab [25]. The higher cross-resistance in previous work could reflect differences between studies in the strains tested, the experimental conditions, or both. The practical impact of the previously observed 23 to 40-fold cross-resistance between Cry1Ab and Cry1F would be greater than that of the only 1.6 to 2.4-fold cross-resistance between these toxins seen here.

Consistent with previous studies [25,27], selection here with either Cry1Ab or Cry1F did not cause cross-resistance to Cry1Ie (Table 1). Conversely, selection here with Cry1Ie caused significant cross-resistance of 2.5-fold to Cry1Ab and >160-fold to Cry1F, whereas previous selection with Cry1Ie yielded >850-fold resistance to Cry1Ie and no cross-resistance to either Cry1Ab or Cry1F [25,27].

The >160-fold cross-resistance to Cry1F caused by selection with Cry1Ie is exceptional, particularly because the amino acid sequence similarity is not high between these toxins: 55% overall and 40% in domain II (Table 2), which is important in toxin binding [14]. In a review evaluating cross-resistance in seven pairs of Bt toxins, >100-fold cross-resistance was associated with >80% amino acid sequence similarity of domain II in nearly all cases [14]. However, the amino acid sequence similarity between Cry1Ie and Cry1F was 70% in domain III (Table 2), which also is involved in toxin binding [14,30]. It remains to be determined if the resistance and cross-resistance seen here are caused by reduced toxin binding or by other mechanisms. Because Cry1F was tested as activated toxin, it is unlikely that reduced proteolytic conversion of protoxin to activated toxin contributes substantially to the high cross-resistance to Cry1F.

The asymmetrical cross-resistance in *O. furnacalis* between Cry1Ie and both Cry1Ab and Cry1F seen here, but not in previous work, implies that the Cry1Ie resistance selected here differs qualitatively from that previously reported. Asymmetrical cross-resistance has also been reported between Cry1Ac and Cry2Ab in pink bollworm [31]. Analogous to a hypothesis proposed to explain this pattern in pink bollworm, the asymmetrical cross-resistance in *O. furnacalis* could occur if resistance to either Cry1Ab or Cry1F requires a resistance allele at only one locus, whereas Cry1Ie resistance requires resistance alleles at two loci, one of which also confers resistance to the other two toxins. If so, then selection with Cry1Ie would cause cross-resistance to Cry1Ab and Cry1F, but selection with Cry1Ab and Cry1F would not cause cross-resistance to Cry1Ie. This hypothesis remains to be tested. The asymmetrical cross-resistance detected in both pink bollworm and *O. furnacalis* confirms the utility of evaluating cross-resistance in both directions, rather than assuming cross-resistance is always symmetrical.

In the field, most Bt crop pyramids are used under conditions somewhere between the ideal scenario evaluated with diamondback moth [15,16] and the nearly worst-case scenario tested here [11,14]. Experimental tests under realistic conditions and retrospective evaluations of pyramids based on field outcomes will improve assessment of their durability relative to the deployment of single-toxin plants and other strategies. Meanwhile, the results of this study imply that under nearly worst-case conditions, pyramids are not necessarily more durable than Bt toxins used singly.

## 4. Materials and Methods

### 4.1. Bt Toxins

Trypsin-activated Cry1Ab and Cry1F toxins (98% purity) used in this study were produced by Marianne P. Carey, Case Western Reserve University, USA. Recombinant Cry1Ie protoxin, expressed in *E. coli*, was purified chromatographically to >92% purity (ABZYMO Biosciences, Beijing, China), by His-tag affinity chromatography on a Ni column [32].

### 4.2. Insects

We used susceptible strain S of *O. furnacalis*, which we started by collecting >1200 diapausing larvae in late September 2012 from non-Bt corn stalks from farmers’ fields in Yangling, Shaanxi Province, China, where Bt corn is not grown. Larvae were kept at ca. 4 °C until April 2013, then moved into the insectary for rearing to pupae at 27 ± 1 °C with 16L:8D and 70 to 80% relative humidity. We obtained 560 pupae and used them to establish the S strain. We used standard methods for rearing *O. furnacalis*, including rearing of larvae on artificial diet [33,34].

### 4.3. Selection Experiment

We started the selection experiment in March 2015, after rearing >6 generations of the S strain on an untreated diet. For each of the five strains tested under a different selection regime, we started with >20,000 neonates (two containers with >10,000 neonates per container). When most survivors had pupated, all pupae were harvested from the two rearing containers and transferred into an oviposition screen cage. Eggs were used to continue each of the five selected strains and for bioassays.

In each of 14 generations, larvae of each of the five selected strains were reared on an artificial diet with Bt toxins incorporated either singly (Cry1Ab, Cry1F, or Cry1Ie) or in pairs (Cry1Ab + Cry1F or Cry1Ab + Cry1Ie). Within China, the number of generations per year for *O. furnacalis* ranges from 1 to 7 [20,35], so the 14 generations studied here represent a range of 2 to 14 years in the field. The concentration (in μg toxin per g diet) used for selection was fixed across generations and was close to the initial LC_50_ value for the S strain (Appendix A): 0.2 for Cry1Ab, 0.5 for Cry1F, and 2.0 for Cry1Ie. At these concentrations, previous work enabled us to estimate that the dominance parameter *h* (which varies from 0 for recessive resistance to 1 for dominant resistance) was >0.8 for each toxin. Thus, the inheritance of resistance was not recessive [25,26,36].

### 4.4. Bioassays

We used diet incorporation bioassays [37] to evaluate larval susceptibility to Cry1Ab, Cry1F, and Cry1Ie. Diet was dispensed into wells of a 48-well plate, which was then infested with 1 neonate per well and held at 27 ± 1 °C with 16L:8D and 70 to 80% relative humidity. Mortality was determined after 7 days. For the strains selected with Cry1Ab alone and the two strains selected with Cry1Ab + Cry1F or Cry1Ab + Cry1Ie, we used bioassays with a range of 4 to 9 concentrations of Cry1Ab to evaluate susceptibility for the generation before selection (0) and each of the next 14 generations (Appendix A). These bioassays each used a range of 5 to 10 concentrations of Cry1Ab (including controls with no toxin), with 48 larvae tested per concentration. We conducted two replicates of each bioassay, each on a different day. For the strains selected with Cry1F or Cry1Ie singly or in pairs with a second toxin, we used the same approach to evaluate susceptibility to these toxins for generations 0 to 5 (Appendix A). However, high levels of resistance to Cry1F and Cry1Ie evolved rapidly. To reduce the expense associated with the large amount of Cry1F and Cry1Ie needed to kill larvae, we tested fewer generations with fewer concentrations of these two toxins from generations 6 to 14 (Appendix A). For the susceptible S strain, which was not selected with Bt toxins, we used 5 to 7 concentrations of each toxin to evaluate susceptibility to Cry1Ab, Cry1F, and Cry1Ie in each of 6 to 7 generations (Appendix A).

### 4.5. Cross-Resistance

To assess cross-resistance, larvae of each of the 3 strains selected with a single toxin were selected for a total of 29 generations (the 14 generations mentioned above plus 15 additional generations) at the fixed concentrations listed above. After 29 generations of selection, we used the bioassay methods described above to test each strain for resistance to the toxin it was selected with and cross-resistance to the other two toxins (Table 1). As the control, we also tested the unselected S strain after 29 generations.

### 4.6. Amino Acid Sequence Similarity

We determined amino acid sequence similarity for pairs of Bt toxins by pairwise sequence comparisons using Alignment of Vector NTI Advance 11 software hosted by ThermoFisher Scientific (https://www.thermofisher.com/cn/zh/home/life-science/cloning/vector-nti-software/vector-nti-advance-software/whats-new-in-vector-nti-advance.html).

### 4.7. Statistical Analysis

We analyzed concentration-mortality data with a probit model using the PoloPlus program to calculate values for LC_50_ with 95% fiducial limits [38]. For the selected strains and the unselected S strain, we used this approach to analyze data for each generation separately. For the unselected S strain, we also used this approach to analyze data pooled from six to seven generations. We calculated the resistance ratio (RR) as the LC_50_ for a selected strain divided by the LC_50_ of for the S strain. To calculate RRs for the selection experiment, we used the LC_50_ for the S strain based on the pooled data. To calculate RRs for cross-resistance after 29 generations of selection, we used the LC_50_ for the S strain based on generation 29. We used PoloPlus to calculate 95% confidence intervals (CIs) for RRs. To determine if differences were statistically significant, we used the conservative criterion of no overlap between 95% fiducial limits for LC_50_ values and no overlap between 95% CIs and 1.0 for RRs. The data used for analyses and Figure 1 are available in Appendix A.

## Figures and Tables

**Figure 1 toxins-11-00461-f001:**
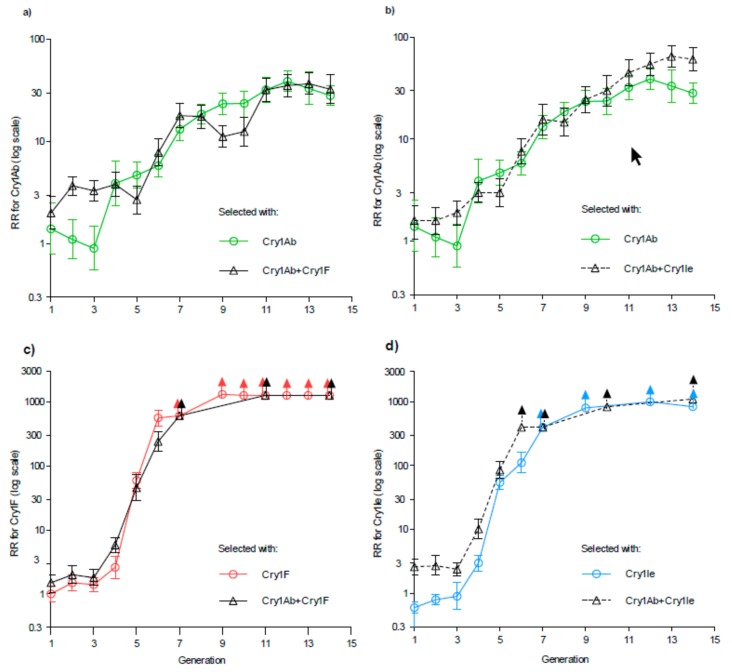
Evolution of resistance to single *Bacillus thuringiensis* (Bt) toxins versus pairs of Bt toxins in *Ostrinia furnacalis*. Strains were selected with (**a**) Cry1Ab alone versus Cry1Ab + Cry1F, (**b**) Cry1Ab alone versus Cry1Ab + Cry11e, (**c**) Cry1F alone versus Cry1Ab + Cry1F, (**d**) Cry1Ie alone versus Cry1Ab + Cry11e. The y-axis shows the resistance ratio (RR), which is the concentration of toxin killing 50% of larvae tested (LC_50_) for a selected strain divided by the LC_50_ of the same toxin for the susceptible parent strain (S) from which the selected strains were derived. Error bars show the 95% confidence limits. Arrows indicate results where the highest concentration tested killed less than 50%, which means the value for RR depicted is the lower limit (see Appendix A).

**Table 1 toxins-11-00461-t001:** Responses of three strains of *Ostrinia furnacalis* to Cry1Ab, Cry1F, and Cry1Ie after 29 generations of selection with a single toxin.

Selected with ^a^	Response to	n ^b^	LC_50_ (95% FL) ^c^ (μg toxin/g diet)	RR (95% CI) ^d^
None (S) ^e^	Cry1Ab	672	0.36 (0.28–0.46)	1.0 (0.7–1.4)
None (S)	Cry1F	672	0.62 (0.46–0.79)	1.0 (0.7–1.5)
None (S)	Cry1Ie	672	5.15 (4.18–6.18)	1.0 (0.8–1.3)
Cry1Ab	Cry1Ab	672	5.30 (3.68–6.88) *	15 (9.9–22) *
Cry1Ab	Cry1F	768	1.00 (0.68–1.36) *	1.6 (1.1–2.5) *
Cry1Ab	Cry1Ie	672	4.64 (3.09–6.29)	0.9 (0.6–1.4)
Cry1F	Cry1Ab	672	0.88 (0.68–1.08) *	2.4 (1.7–3.4) *
Cry1F	Cry1F	96	>1000	>1600
Cry1F	Cry1Ie	672	4.16 (3.24–5.22)	0.8 (0.6–1.1)
Cry1Ie	Cry1Ab	768	0.92 (0.77–1.12) *	2.5 (1.9–3.5) *
Cry1Ie	Cry1F	672	>100	>160
Cry1Ie	Cry1Ie	96	>2047	>840

^a^ Single toxin used to select each strain. ^b^ Number of larvae tested in bioassays. ^c^ Concentration of toxin killing 50% of larvae and its 95% fiducial limits. ^d^ Resistance ratio and its 95% confidence interval. ^e^ Susceptible S strain was reared without exposure to any Bt toxin. * Significantly higher than the S strain by non-overlap of the 95% fiducial limits of the LC_50_ and lower limit of the 95% confidence interval of the RR > 1.

**Table 2 toxins-11-00461-t002:** Amino acid sequence similarity for pairs of *Bacillus thuringiensis* (Bt) toxins.

		Amino Acid Sequence Similarity (%)			
Toxin Pair		Domain I	Domain II	Domain III	Overall
Cry1Ab	Cry1F	74	50	63	63
Cry1Ab	Cry1Ie	62	44	80	59
Cry1F	Cry1Ie	62	40	70	55

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
