# Peer review of "Evolution of Asian Corn Borer Resistance to Bt Toxins Used Singly or in Pairs"

_toxins, 2019, doi:10.3390/toxins11080461_

Round 1

Reviewer 1 Report

The authors are trying to determine the evolution of Bt resistance between single proteins and pyramids against Asian corn borer under a suboptimal toxicity regime as sort of a "worst case" scenario.  the authors argue that this "worst case" scenario has only been modeled but never demonstrated (this reviewer disagrees and will discuss later).  The authors have conducted a considerable amount of research for this manuscript although this reviewer finds little value in this research as the results are expected and of little “real world” conditions as written

The authors need to address the following questions/comments:

1)  When would there ever be a Bt corn product against Asian corn borer that would only control 50% of the population (for one or two proteins) at commercial launch when current products should control >99% of the population?  So as currently written, this hypothesis has little to no field value or relevance

The primary example this reviewer could think of where a product would only have 50% mortality is when there already was a considerable amount of resistance to at least one of the proteins.  Note that this has been demonstrated in the field several times and have been discussed in the context in the potential weakness of a pyramid:  Cry3Bb1 and Cry34/35 against western corn rootworm, and Cry1Ac and Cry2Ab against pink bollworm.  The authors should cite these examples and put their example in this context.  But if there already is resistance to one of the proteins, and there is at least some cross resistance to others (at least in some cases or when the resistance mechanism is more general), then resistance to the pyramid should increase quickly

2.  The authors state that inheritance (h) to these proteins at the concentrations tested was >0.8 for each of all three of the proteins.  At this level, resistance would most likely NOT be confirmed by an alteration in binding, and that several of these other resistance mechanism may be more general in their nature conferring cross resistance to other proteins.  The authors need to address this in their introduction and discussion

3.  The authors show in Table 1 that there was significant cross resistance between Cry1Ie (selected0 and Cry1F.  Is this why the authors did not show data for this pyramid in Figure 1? (or is “D” in figure 1 mslabelled?  If “D” is Cry1Ie and Cry1F, is it surprising to see similar RR’s for the pyramid vs the single?  So with this relatively high level of cross resistance between Cry1Ie and Cry1F, was this an appropriate choice for including in this study?  Why not Cry2Ab2 or Vip3A or some other protein?

4. (related to #2).  So if this reviewer understands the methods, the authors are selecting at 50% mortality for single proteins and roughly 85% or so for the pyramid, correct?  So even though the selection pressure is different, this relatively low level of selection pressure would not only predict high levels of resistance (a considerable number of heterozygotes are still available for mating producing homozygous resistant insects for resistance that is recessive, or a lot of heterozygotes that are functionally resistance) but that the resistance mechanism selected for could be a number of things (not just binding), and that several of these mechanism such as proteolysis could be more general so that even though you are selecting for resistance with 2 different proteins, functionally you are selecting only with one.

Lines 33-34:  The authors state that the pyramid strategy has limited experimental data and do not cite any experimental data.  This review suggests that the authors visit the scenario of controlling European corn borer (Ostrinia nubilalis) with Cry1A.105, Cry1F and Cry2Ab2 pyramided corn.

Lines 51-53.  The authors state that they are using Bt toxins in artificial diet to mimic Bt plants.  Besides the fact that Bt plants would almost certainly not be commercialized to only control 50% of the population, the authors need to comment on differences in toxicity and insect response between using artificial diet bioassays and plants.

Lines 115-118.  Yes, there was significant cross resistance based on your LC50 results.  However, please put a RR of 1.6 fold and 2.4 fold of cross resistance in the context of comparing against insects that are >800-fold resistant to an individual protein.

Lines 151-157 and Table 2:  As discussed above, the resistant mechanism(s) in this study most likely are not binding based (if inheritance is >0.8).  This reviewer suggests that the authors need to discuss potential other mechanisms of resistance, or present an argument why the resistance mechanism could be binding even though inheritance is quite dominant.

Lines 180-183:  As Cry1Ie is not very well known or characterized, this reviewer suggests to include the source of the protein.

Author Response

Point-by-Point Reply

The authors are trying to determine the evolution of Bt resistance between single proteins and pyramids against Asian corn borer under a suboptimal toxicity regime as sort of a "worst case" scenario.  the authors argue that this "worst case" scenario has only been modeled but never demonstrated (this reviewer disagrees and will discuss later).  The authors have conducted a considerable amount of research for this manuscript although this reviewer finds little value in this research as the results are expected and of little “real world” conditions as written.

Reply: We appreciate and agree with the comment that this paper reports the results from a considerable amount of research.

Please note: Web of Science lists over 400 citations for the two papers describing experimental tests of the pyramid strategy using a model system with noncommercial Bt broccoli vs. diamondback moth [references 15 and 16]. These papers have had tremendous influence despite the fact little or no connection exists in many cases between the ideal conditions studied and “real world” conditions. The submitted paper focusing on nearly worst-case conditions is useful for bringing some balance to this issue.

Please see the point-by-point reply to each specific comment below.

1. When would there ever be a Bt corn product against Asian corn borer that would only control 50% of the population (for one or two proteins) at commercial launch when current products should control >99% of the population?  So as currently written, this hypothesis has little to no field value or relevance. 

The primary example this reviewer could think of where a product would only have 50% mortality is when there already was a considerable amount of resistance to at least one of the proteins.  Note that this has been demonstrated in the field several times and have been discussed in the context in the potential weakness of a pyramid: Cry3Bb1 and Cry34/35 against western corn rootworm, and Cry1Ac and Cry2Ab against pink bollworm.  The authors should cite these examples and put their example in this context.  But if there already is resistance to one of the proteins, and there is at least some cross resistance to others (at least in some cases or when the resistance mechanism is more general), then resistance to the pyramid should increase quickly.

Reply: Revised as suggested:

Introduction (L48-51):

“When Bt toxins are first used in single-toxin crops, they usually cause high mortality of susceptible target pests. However, pyramids are often deployed after field-evolved resistance has decreased the efficacy of one or more toxins in the pyramid, so the mortality caused by each toxin alone may be 50% or less [9-11].”

The three references cited [9-11] review both examples mentioned above as well as examples in other pests including Helicoverpa zea.

2.  The authors state that inheritance (h) to these proteins at the concentrations tested was >0.8 for each of all three of the proteins.  At this level, resistance would most likely NOT be confirmed by an alteration in binding, and that several of these other resistance mechanism may be more general in their nature conferring cross resistance to other proteins.  The authors need to address this in their introduction and discussion.

Reply: Revised as suggested (Discussion, L182-185):

“It remains to be determined if the resistance and cross-resistance seen here are caused by reduced toxin binding or by other mechanisms. Because Cry1F was tested as activated toxin, it is unlikely that reduced proteolytic conversion of protoxin to activated toxin contributes substantially to the high cross-resistance to Cry1F.”

 3.  The authors show in Table 1 that there was significant cross resistance between Cry1Ie (selected0 and Cry1F.  Is this why the authors did not show data for this pyramid in Figure 1? (or is “D” in figure 1 mslabelled?  If “D” is Cry1Ie and Cry1F, is it surprising to see similar RR’s for the pyramid vs the single?  So with this relatively high level of cross resistance between Cry1Ie and Cry1F, was this an appropriate choice for including in this study?  Why not Cry2Ab2 or Vip3A or some other protein? 

Reply: Revised as suggested. Thanks for catching the error in the legend for Figure 1d. We tested Cry1Ab + Cry1Ie (not Cry1F and Cry1Ie) and revised accordingly:

L109. Fig. 1d legend: “d) Cry1Ie alone versus Cry1Ab + Cry11e.”

4. (related to #2).  So if this reviewer understands the methods, the authors are selecting at 50% mortality for single proteins and roughly 85% or so for the pyramid, correct?  So even though the selection pressure is different, this relatively low level of selection pressure would not only predict high levels of resistance (a considerable number of heterozygotes are still available for mating producing homozygous resistant insects for resistance that is recessive, or a lot of heterozygotes that are functionally resistance) but that the resistance mechanism selected for could be a number of things (not just binding), and that several of these mechanism such as proteolysis could be more general so that even though you are selecting for resistance with 2 different proteins, functionally you are selecting only with one.

Reply: Revised as suggested. Please see reply to comment #2 above.

5. Lines 33-34:  The authors state that the pyramid strategy has limited experimental data and do not cite any experimental data.  This review suggests that the authors visit the scenario of controlling European corn borer (Ostrinia nubilalis) with Cry1A.105, Cry1F and Cry2Ab2 pyramided corn.

Reply: Revised as suggested. We revised to clarify (L36-38):

“Although the pyramid strategy has been studied extensively with theoretical models, experimental data from selection experiments testing toxins singly and in combinations are limited [11-16]. Moreover, previous selection experiments have focused on ideal or nearly ideal scenarios, which may not necessarily reflect field conditions.”

References 11-14 cited above review the relevant experimental data. References 15 and 16 report the experimental data most often cited in support of the pyramid strategy, as explained in the next sentence:

“The primary empirical evidence supporting use of the pyramid strategy for Bt crops is from a model system with diamondback moth and noncommercial Bt broccoli plants that produce Bt toxins Cry1Ac and Cry1C [15-16].”

6. Lines 51-53.  The authors state that they are using Bt toxins in artificial diet to mimic Bt plants.  Besides the fact that Bt plants would almost certainly not be commercialized to only control 50% of the population, the authors need to comment on differences in toxicity and insect response between using artificial diet bioassays and plants.

Reply: Revised as suggested. We omitted the claim that we used Bt toxins in artificial diet to mimic Bt plants (L59):

“We tested Bt toxins Cry1Ab, Cry1F, and Cry1Ie alone and in pairs (Cry1Ab + Cry1F and Cry1Ab + Cry1Ie) incorporated in artificial diet.”

6. Lines 115-118.  Yes, there was significant cross resistance based on your LC50 results.  However, please put a RR of 1.6 fold and 2.4 fold of cross resistance in the context of comparing against insects that are >800-fold resistant to an individual protein.

Reply: Revised as suggested (Discussion, L160-162):

“The significant, symmetrical cross-resistance between Cry1Ab and Cry1F seen here of 1.6- and 2.4-fold is low relative to the 15-fold resistance to Cry1Ab caused by selection with Cry1Ab and the >1600-fold resistance to Cry1F caused by selection with Cry1F (Table 1).”

7. Lines 151-157 and Table 2:  As discussed above, the resistant mechanism(s) in this study most likely are not binding based (if inheritance is >0.8).  This reviewer suggests that the authors need to discuss potential other mechanisms of resistance, or present an argument why the resistance mechanism could be binding even though inheritance is quite dominant.

Reply: Revised as suggested. Please see reply to comment #2 above.

8. Lines 180-183:  As Cry1Ie is not very well known or characterized, this reviewer suggests to include the source of the protein.

Reply: As stated in the Methods (L209-211): “Recombinant Cry1Ie protoxin, expressed in E. coli, was purified chromatographically to >92% purity (ABZYMO Biosciences, Beijing, China), by His-tag affinity chromatography on a Ni column [32].”

The cited reference [32] provides additional information.

Reviewer 2 Report

The authors describe a set of selection experiments to understand how mixtures of proteins, as surrogates for pyramided Bt crops, may or may not slow resistance development under conditions that violate the assumptions of pyramid approaches to resistance management.  The experiments are well conducted and well reported.  However, I have a few suggestions for improvement below and attach a marked up version of the manuscript with specific changes that are needed.

First, the selection regime, using the LC50 for each protein, is unrealistic in representing what may occur in the field.  The events that produce Cry1F and Cry1Ab (TC1507, Bt11, MON810) are at or very close to "high dose" against Asian corn borer, meaning that they exceed the LC99 for each protein.  Therefore, while the authors correctly describe their system as "sub-optimal" for pyramiding, it is actually unrepresentative of any pyramids currently available for Asian corn borer.  The authors need to make this clear, and not claim that there results have implications for current products against this pest.

Second, describing their set-up as "suboptimal" for pyramiding is a vast understatement.  Indeed, at one point they describe the set up as a "nearly worst-case scenario", which is a much fairer description.  The abstract and conclusions should be modified to make it clear that this is not just suboptimal, but worst case that does not reflect the current field situation.

Third, the authors observed low-level cross-resistance among the proteins in their study.  It would be valuable for the reader if they were to discuss how low-level cross-resistance impacts the effectiveness of pyramids, especially for pyramids that are at or close to high dose. This reviewer believes that the cross-resistance measured is too low to impact survival on a high-dose crop situation, and therefore does not undermine the pyramids of these proteins.  Nevertheless, the selection regime used (LC50) was not capable of selecting for high levels of resistance as it allows a significant portion of susceptible insects to move forward across generations.  Results may differ if a higher level of selection was placed on the insect populations.

Finally, it would have been helpful if the authors had run a parallel study using more optimal conditions for pyramid-based resistance management.  As they pointed out in the introduction, there are not many empirical validations of the pyramid approach and they have a nice system to do that, and perhaps evaluate the impact of individual assumptions on the effectiveness of pyramiding.  I recognize that this goes beyond the scope of the present manuscript, but encourage a more comprehensive study in the future.

Author Response

1. The authors describe a set of selection experiments to understand how mixtures of proteins, as surrogates for pyramided Bt crops, may or may not slow resistance development under conditions that violate the assumptions of pyramid approaches to resistance management.  The experiments are well conducted and well reported.  However, I have a few suggestions for improvement below and attach a marked up version of the manuscript with specific changes that are needed.

Reply: We appreciate the positive feedback.

2. First, the selection regime, using the LC50 for each protein, is unrealistic in representing what may occur in the field.  The events that produce Cry1F and Cry1Ab (TC1507, Bt11, MON810) are at or very close to "high dose" against Asian corn borer, meaning that they exceed the LC99 for each protein.  Therefore, while the authors correctly describe their system as "sub-optimal" for pyramiding, it is actually unrepresentative of any pyramids currently available for Asian corn borer.  The authors need to make this clear, and not claim that there results have implications for current products against this pest.

Reply: Revised as suggested.

Abstract (L12-14): “The conditions examined here in a laboratory selection experiment with artificial diet may reflect some aspects of field situations, but do not necessarily represent conditions for currently available Bt corn products against this pest.”

Introduction (L70-71):These unfavorable conditions may reflect some aspects of field situations, but do not necessarily represent conditions for currently available Bt corn products against this pest.”

3. Second, describing their set-up as "suboptimal" for pyramiding is a vast understatement.  Indeed, at one point they describe the set up as a "nearly worst-case scenario", which is a much fairer description.  The abstract and conclusions should be modified to make it clear that this is not just suboptimal, but worst case that does not reflect the current field situation.

Reply: Revised as suggested. We revised “suboptimal" to "nearly worst-case scenario" in the abstract (L9) and throughout the paper.

4. Third, the authors observed low-level cross-resistance among the proteins in their study.  It would be valuable for the reader if they were to discuss how low-level cross-resistance impacts the effectiveness of pyramids, especially for pyramids that are at or close to high dose. This reviewer believes that the cross-resistance measured is too low to impact survival on a high-dose crop situation, and therefore does not undermine the pyramids of these proteins.  Nevertheless, the selection regime used (LC50) was not capable of selecting for high levels of resistance as it allows a significant portion of susceptible insects to move forward across generations.  Results may differ if a higher level of selection was placed on the insect populations.

Reply: Revised as suggested (Discussion, L160-170):

“The significant, symmetrical cross-resistance between Cry1Ab and Cry1F seen here of 1.6- and 2.4-fold is low relative to the 15-fold resistance to Cry1Ab caused by selection with Cry1Ab and the >1600-fold resistance to Cry1F caused by selection with Cry1F (Table 1). In addition, the cross-resistance between Cry1Ab and Cry1F in this study is somewhat lower than in previous studies of O. furnacalis. In previous work, selection with Cry1Ab yielded 40-fold resistance to Cry1Ab and 6-fold cross-resistance to Cry1F [27]. Previous selection with Cry1F yielded >1700-fold resistance to Cry1F and 23-fold cross-resistance to Cry1Ab [25]. The higher cross-resistance in previous work could reflect differences between studies in the strains tested, the experimental conditions, or both. The practical impact of the previously observed 23- to 40-fold cross-resistance between Cry1Ab and Cry1F would be greater than that of the only 1.6- to 2.4-fold cross-resistance between these toxins seen here.”

5. Finally, it would have been helpful if the authors had run a parallel study using more optimal conditions for pyramid-based resistance management.  As they pointed out in the introduction, there are not many empirical validations of the pyramid approach and they have a nice system to do that, and perhaps evaluate the impact of individual assumptions on the effectiveness of pyramiding.  I recognize that this goes beyond the scope of the present manuscript, but encourage a more comprehensive study in the future.

Reply: We agree and appreciate this constructive suggestion.

Comments of noted on pdf

6. L9. This is better described as "nearly worst case conditions" at line 173.  As written, this is misleading.

Reply: Revised as suggested to “nearly worst-case conditions.”

7. L27. "Increasingly rapid" is misleading. It is more that as commercial use of the traits matures there are more incidents of resistance

Reply: Revised as suggested. Omitted “increasingly rapid.”

8. L53. This is misleading.  TC1507, Bt11, and MON810 are all at or very close to high dose and do meet condition (1) above.  The authors should delete this claim.

Reply: Revised as suggested. Omitted “at levels to mimic single-toxin and pyramided Bt plants, respectively.”

9. L59-60. This is lower than in Bt plants (see comment on line 53)

Reply: Revised as suggested. See reply above to comment #8 about line 53.

10. L109. It would be more faithful to the data to stop the curves in the figures after 6 and 5 generations respectively.

Reply: We prefer to report all of the data for LC50 values. The limitations of these data after 6 generations for Cry1F and 5 generations for Cry1Ie are stated clearly in the text.

11. Table 1. So resistance ratio did not increase after generation 14 (even declined, perhaps?)

Reply: Yes.

12. Table 1. looks like a data entry error (these don't align with the LC50 for the S strain on Cry1Ie)

Reply: Thanks for catching this error! We revised the last line of Table 1 to correct this: LC50 >2047 and resistance ratio >840.

13. L142. These ratios are very small compared with resistance ratios.  You should discuss the biological relevance of these low levels of cross-resistance relative to the ideal pyramid.  You should also discuss why the cross-resistance ratios you found differ from those cited below

Reply: Revised as suggested. Please see reply above to comment #4.

14. L168. An alternative explanation to discuss would be that the selection with Cry1Ie selected for a mechanism of resistance unrelated to binding - perhaps protein detoxification.

Reply: Revised as suggested (Discussion, L182-184):

“It remains to be determined if the resistance and cross-resistance seen here are caused by reduced toxin binding or by other mechanisms.”

15. L176. "nearly worst case", as worded above (line 173), is a more accurate and fair representation of your conclusions

Reply: Revised as suggested. We revised “suboptimal" to "nearly worst-case."

16. L180. How closely related are these toxins to those produced by events TC1507, Mon810 and Bt11?  Has their biological equivalence been established?

Reply: We have omitted the claim that our experiments mimic Bt corn, which diminishes the relevance of the questions above to our paper. Nonetheless, the amino acid sequences were similar for the Cry1F we used and Cry1F in TC1507, likewise for Cry1Ab we used and Cry1Ab in Mon810 and Bt11. Thus, similar biological activity is expected.

17. L204. This makes sense at the LC50, but again, at field-relevant concentrations of these proteins, resistance has been shown to be recessive or nearly recessive in Ostrinia.

Reply: The use of the genus name Ostrinia without a species name above suggests the comment may refer to Ostrinia nubilalis. The relevant data for inheritance of resistance to Bt toxins in the pest studied here, Ostrinia furnacalis, are cited in the paper (L234, references 25, 26 and 36) and do not show recessive resistance in this pest. To clarify, we added this sentence (Introduction, L65-67):

“Moreover, in field efficacy tests, Bt corn producing Cry1Ab, Cry1F or Cry1Ie alone or Cry1Ie + Cry1Ac (which is similar to Cry1Ab) provided significant protection, but did not meet the high-dose standard against O. furnacalis [10,23,29].”

18. Methods. With only 4 concentrations (one of which is the control), LC50 estimates will have a great deal of uncertainty and even difficulties in calculating FLs in POLO.  This is not reflected in the 95% FLs in table S2, so I encourage the authors to re-examine the statistical analysis.

Reply: Thanks for catching our error. We have revised the Methods (L242) to correctly report the minimum number of concentrations was five.

19. L230. How did you obtain the amino acid sequences for the toxins used in this study?

Reply: From the Crickmore Bt toxin website.

Round 2

Reviewer 1 Report

The authors are trying to determine the evolution of Bt resistance between single proteins and pyramids against Asian corn borer under a suboptimal toxicity regime as sort of a "nearly worst case" scenario.  the authors argue that this "nearly worst case" scenario has only been modeled but never demonstrated. This is a revised manuscript and the authors have addressed this reviewer’s comments for the most part. This reviewer appreciates that the authors highlighted there edits in the text. The authors have conducted a considerable amount of research for this manuscript although this reviewer finds little value in this research as the results are expected and of little “real world” conditions as written

This reviewer would like to make one comment regarding the author’s response below

From the authors: “Please note: Web of Science lists over 400 citations for the two papers describing experimental tests of the pyramid strategy using a model system with noncommercial Bt broccoli vs. diamondback moth [references 15 and 16]. These papers have had tremendous influence despite the fact little or no connection exists in many cases between the ideal conditions studied and “real world” conditions. The submitted paper focusing on nearly worst-case conditions is useful for bringing some balance to this issue”. Reviewer comments: This reviewer is glad that the authors include these citations regarding DBM and broccoli.  And yes, these manuscripts have had considerable importance in the Bt resistance field over the last 20 years ago since they were published. But these manuscripts were published at the beginning of the transgenic plant revolution when we needed this type of data, not at least 20 years later when we have plenty of modeling and empirical data to draw from

Abstract:

Line 5:  Remove “rapid”.  To many scientists, Bt resistance occurred as expected, and was not “rapid”.

Line 7:  Delete or change.  As discussed in this reviewers original response, there is considerable empirical data now available.

Lines 11-14. This reviewer thanks the authors for considering this comment.  But this reviewer questions if this comment belongs in the Abstract (perhaps better in Discussion).  Keeping this comment in the Abstract would most likely dilute the importance of this manuscript.

Introduction

Lines 35-38: The authors may be correct here for experimental testing, but this reviewer would argue that we now have plenty of real-world field examples

Line 39:  Please include:  “The primary empirical experimental evidence…”

Line 47: “…they usually cause relatively high mortality…”.  Note:  It would be nice for the authors to describe numerically what they mean by “high “ mortality.  Also note: that one case for pyramids not performing as well as hoped is with SmartStax that is a pyramid with Cry3Bb1 and Cry34/35.  In this case, even at the beginning, each protein had 95-99% mortality.  Is this considered “high” by the authors?

Line 48-49:  Please change to: “…after the efficacy of one or more toxins in the pyramid has decreased..”.  (one can get reduced efficacy before we would state “field-evolved resistance”).  Note:  From this reviewer, although there may be pyramids in which one of the proteins has 50% or less efficacy due to resistance, this reviewer doubts that this pyramid would be given the same durability assumptions as a protein with greater efficacy.  Also note that there is at least one example for lepidoptera where a pyramid was registered against a particular pest, when only one of the two proteins had substantive toxicity against the targeted pest.

Discussion:

Line 160.  Please insert a sentence stating that cross resistance in Ostrinia between Cry1Ab and Cry1F is not unexpected due to shared binding sites (Hernandez-Rodriguez, 2013, PLOS ONE)

Materials and Methods:

Lines 207-209.  Probably the only original request by this reviewer that the authors did not adequately change.  As Cry1Ie is relatively unknown, this reviewer would request most information about this protein other than just a citation.
